# Nature-based interventions for individual, collective and planetary wellbeing: A protocol for a scoping review

**Jesse Blackburn**, **Afonso Pereira**, **Luke Jefferies**, **Andrew H. Kemp**, **Amy Isham**\*

School of Psychology, Faculty of Medicine, Health, and Life Science, Swansea University, Swansea, United Kingdom

\* a.m.isham@swansea.ac.uk

## Abstract

Nature-based interventions (NBIs) provide an opportunity to enhance individual wellbeing, improve community cohesion, and promote a culture of care for the environment. Several scoping reviews have attempted to catalogue the positive effects of NBIs on wellbeing, yet, these have typically focused on outcomes relating to individual wellbeing, thus restricting the assessment of the possible benefits of NBIs. Here we present a protocol for a scoping review that will synthesise the evidence relating to the impact of NBIs across a much broader range of domains with a focus on self (individual wellbeing), others (collective wellbeing) and nature (planetary wellbeing). This scoping review will also provide insight into the relative effectiveness of different types of NBIs at enhancing wellbeing across these domains and synthesise the underlying theory on which interventions have been developed and reported outcomes have been presented. A literature search for theses and peer-reviewed studies will be conducted on four databases (APAPsycINFO, Web of Science, Medline, and Scopus) and ProQuest Dissertations & Theses Global. Two independent reviewers will complete a two-stage screening process (title/abstract and full-text) using the Covidence platform. The protocol for this scoping review is registered with the Open Science Framework. Data extraction will focus on publication details, type of intervention, and wellbeing-related outcomes. Results will be reported in a scoping review following standardised guidelines relating to the Preferred Reporting Items for Systematic Reviews and Meta-Analyses Extension for Scoping Reviews. This research will inform the design and delivery of NBI's across a wide range of sectors including health and social care, public policy, education, and community services, to ultimately promote human flourishing at scale.

## Introduction

The term 'nature-based intervention' (NBI) refers to a diverse range of interventions which aim to use nature exposure to enhance human health and wellbeing [1]. There is much opportunity to align this work to also promote planetary wellbeing, which was recently defined as the 'highest attainable standard of wellbeing for human and non-human beings and their social and natural

**Data availability statement:** No datasets were generated or analysed during the current study. All relevant data from this study will be made available upon study completion.

**Funding:** JB's time is funded through a PhD studentship awarded by Swansea University's Faculty of Medicine, Health, and Life Science. The funder had no role to play in the study design, data collection and analysis, decision to publish, or preparation of the manuscript.

**Competing interests:** The authors have declared that no competing interests exist.

systems' [2, p. 4]. Previous reviews have explored the link between NBIs and wellbeing [3–5], yet these reviews have typically focused on narrow, individualistic conceptions of wellbeing. For instance, outcomes have focused on how exposure to green spaces such as parks and gardens, is associated with improved mood and emotion [6,7], enhanced cognitive function [8], and a reduction in stress and stress-related illnesses [9,10]. This approach strips away important social ecological context, isolating the individual from community and nature [11,12]. It also reinforces neoliberal ideologies [13], which characterise mental health difficulties as those arising from individual dysfunction, disregarding social and environmental factors. Consequently, wellbeing interventions have been designed to enhance wellbeing at the level of the individual, while social and environmental aspects of wellbeing go unaddressed. We suggest that NBIs have the potential to serve as a more holistic wellbeing intervention that positively influences the individual, social, and environmental aspects of wellbeing. Accordingly, the aim of this review is to synthesise the evidence relating to the impacts of NBIs across a broad range of wellbeing domains, providing a more holistic understanding.

## What are nature-based interventions?

A common feature of NBIs is the use of natural environments to improve health and wellbeing across a range of contexts including healthcare [4,14,15], education [16,17], and community-based settings [18]. The mode of delivery for NBIs is highly heterogenous, varying by the type of nature [19], duration of exposure [20], level of interaction [21], intensity of physical activity [22], and presence of other people [23] or non-human animals [16,24].

In an umbrella review on NBIs, Harper et al. [25] created a list of five distinct categories of nature-based therapies based on past work, which included nature-based therapy, forest therapy, horticultural therapy, wilderness therapy, and adventure therapy. Whilst informative, this review was conducted within the context of clinician-led therapies and thus was largely constrained to measurements of individual wellbeing, such as addressing substance abuse, ill health, and cognitive performance. Consequently, Harper et al.'s review [25] excluded non-clinician-led studies and investigations with a broader focus including indicators of collective and planetary wellbeing, concepts that will be introduced below. Nonetheless, these five categories may be helpfully applied to the broader scope of interest identified in this protocol. The term 'intervention,' rather than 'therapy' will be adopted for the purposes of this review, whereby 'intervention' may be understood as action taken towards either ameliorating an illness, dysfunction, or unwanted symptoms, or otherwise promoting wellbeing and positive psychological experience [1,26,27]. We consider 'intervention' to be a broader term that does not necessitate direct interaction with a therapist, and therefore will allow the review to capture impacts of therapy-based as well as non-therapy-based NBIs. Thus, the altered categories presented by Harper et al. [25] will serve as a framework for categorising NBIs identified in the review process.

## What is wellbeing?

Dictionary definitions of wellbeing focus on the state of being comfortable, healthy, and happy. Dominant scholarly accounts of wellbeing also emphasise the immediate experience of the individual, focusing on constructs such as life satisfaction [e.g., 28] or happiness [e.g., 29], often to the exclusion of important system-wide contextual factors such as social relationships and the health of the non-human environment [13,30,31, but see 32]. Recent developments in the field characterise wellbeing more holistically, with a much wider focus that includes the self, others, and nature [e.g., 13,33]. These developments highlight individual, collective, and even planetary wellbeing, inspired by social ecological models [34,35]. In this regard,

individual wellbeing has been defined as the capacity to feel good and function well, exemplified by positive emotions, physical health, and achieving personal goals [36]. Collective wellbeing has been described as feeling good and functioning well, interdependently, signified, for example, by high group cohesion, mutual support, and a sense of belonging [37]. The more recently coined phrase 'planetary wellbeing' has been defined as 'the highest attainable standard of wellbeing for human and non-human beings and their social and natural systems' [2], p. 4. Encompassing pro-environmental behaviours, such as reducing waste and conserving biodiversity, planetary wellbeing is a systems-oriented perspective that integrates ecosystem health and human wellbeing. The JYU.Wisdom Community [38] defines planetary wellbeing as a state in which the Earth's natural systems are able to sustain life whilst simultaneously fulfilling organisms' needs, allowing them to actualise their inherent capacities.

Researchers have typically focused on these different aspects of wellbeing in parallel. The recently developed GENIAL framework [11,13,33,39,40] defines wellbeing from a biopsychosocial-ecological perspective, encompassing the individual, community, and natural environment. By building upon concepts presented in biopsychosocial models [41], ecological systems theory [34,35], and sustainable wellbeing [13], GENIAL integrates evidence from diverse fields – including positive psychology, epidemiology, and environmental health – to provide a unified, holistic understanding of wellbeing that recognises the interdependence of individual, collective, and planetary domains of wellbeing [40].

This model has encouraged the development of innovative interventions, such as a community-supported agriculture initiative to support the wellbeing of local communities through the lens of sustainable food (Pereira, Blackburn et al., [unpublished]). This and related work [e.g., 12,39] characterise what has been described as an emerging third wave of positive psychology, underpinned by an 'epistemological broadening' [42, p. 660] that spans beyond the individual and incorporates broader and system-wide contextual factors, including the groups, cultures, and systems in which people are embedded [42–45]. This broadening draws attention to the interdependence of ecological systems and human wellbeing [12], acknowledging that environmental challenges are also challenges to human wellbeing [46].

Recognising wellbeing as a multidimensional construct provides a clear attempt to move beyond the concept of wellbeing as a 'wicked problem', an intractable issue that allows for neither simple definitions nor clear (and reproducible) solutions. Part of the 'wickedness' of wellbeing lies in often-contradictory approaches to addressing wellbeing at different levels of scale and the intersectionality of complex societal problems, including inequalities and climate change. Actions aimed at enhancing wellbeing within one domain may have system-wide effects that either facilitate or inhibit wellbeing in other domains. Acknowledging the systematic interaction of problems—and potential solutions—across a diversity of domains is essential when attempting to resolve wicked problems [47,48]. We suggest that Nature-Based Interventions (NBIs) may be uniquely placed to influence wellbeing across individual, social, and environmental domains, whilst GENIAL provides a holistic framework well-suited to conceptualising and assessing the reciprocal and cumulative effects of promoting wellbeing at multiple levels.

## So why is another review on NBIs needed?

Several reviews on NBIs have now been published [1,49–51], as have multiple 'reviews of reviews' [25,52,53, see also 54]. Yet, previous reviews have failed to investigate the effects of NBIs across multiple domains relevant to a more holistic understanding of wellbeing. Past reviews have largely focused on aspects relating to individual wellbeing (and 'illbeing'), while a focus on collective or planetary wellbeing is less frequently reported.

For instance, the meta-review published by Antonelli et al. [52] identified 16 reviews that investigated the effects of forest bathing on wellbeing. Of these, only a single review reported on a measure of social wellbeing [55], and no reviews have yet focused on the construct of planetary wellbeing, despite the proliferation of research investigating the effects of human connection to - and interdependence with - nature and the implications such connection has for pro-environmental attitudes and behaviours [56–59].

Our focus will be a scoping review, an increasingly popular method of systematic literature review [60–62] that provides the opportunity to identify the volume and characteristics of primary research, including key themes and knowledge gaps, in the subject area of interest [62,63]. Previously published scoping reviews on NBIs have focused on a diversity of issues such as mental health [64], bereavement care [65], disease [66,67], and health-related behaviours [5]. There are also several published protocols for scoping reviews that intend to focus on impacts of NBIs on environmental behaviour [68], and parent-child relationships [69]. In a scoping review on NBIs for vulnerable youth [16], the majority of original studies (N = 68) reported on measures of individual and social outcomes, yet none of the studies provided a measure relating to planetary health such as changes in pro-environmental behaviours (e.g., waste reduction and recycling). In contrast, a systematic review presented by Silva et al. [51] included 38 studies where NBIs were used to enhance wellbeing and nature-related outcomes (e.g., nature affinity). Silva et al. [51] identified only four studies which provided a measure of social wellbeing [70–73]. Several factors may have contributed to the limited number of social outcomes observed in this review. Notably, Silva et al. [51] employed only a single search term related to social outcomes, 'social,' selected after a brief review of the literature. This narrow selection of search terms might have restricted the scope of their findings.

The present scoping review aims to address this limitation by incorporating a broader and more diverse set of search terms related to NBIs and wellbeing, particularly social wellbeing. For instance, while Silva et al. [51] used only the term 'social,' our review includes additional related terms such as 'loneliness,' 'affiliation,' 'community,' 'belonging,' and 'neighbourhoods.' Our review also offers a greater variety of terms relating to different types of NBIs, such as forest therapy, horticultural therapy, and animal-assisted therapies in outdoor environments. This approach is intended to capture a wider array of relevant studies, thereby providing a more comprehensive synthesis of the available evidence.

Moreover, Silva et al. [51] excluded studies involving virtual reality, participants with serious mental health conditions or severe physical disorders, and self-guidance (i.e., without researcher or practitioner presence). These exclusions might have further limited the scope of their review. By contrast, our review encompasses a broader range of studies, including those with virtual reality components and diverse participant profiles, to ensure a more inclusive and comprehensive understanding of holistic wellbeing outcomes. Finally, Silva et al.'s review included articles published until July 2020 [51]. The present review, therefore, offers the opportunity to include more recent literature in what is a rapidly growing field of investigation. By incorporating studies published after this date, our review aims to provide an up-to-date synthesis of current evidence.

Collectively, previously published reviews, including the work of Silva et al. [51], have demonstrated a great variety of evidence relating to NBIs. However, no single review has comprehensively synthesised the available research on the effects of NBIs on an epistemologically broadened conceptualization of wellbeing that includes a focus on self, others and nature. This scoping review aims to build on Silva et al.'s epistemological breadth by incorporating a more extensive set of search terms, broader inclusion criteria, and more recent literature. In doing so, we will advance understanding of the extent to which different types of NBIs influence

wellbeing-related outcomes across (1) individual, (2) collective, and (3) planetary domains, and identify current gaps in knowledge.

## Methods

The proposed scoping review, which is expected to take approximately 12 months to complete, will commence October 2024 (see Table 1) and is informed by the Joanna Briggs Institute (JBI) methodology for scoping reviews [62] and the Preferred Reporting Items for Systematic Reviews and Meta-Analyses Extension for Scoping Reviews [PRISMA-ScR; 74]. The protocol of this scoping review was developed according to the Preferred Reporting Items for Systematic Review and Meta-Analysis Protocols [PRISMA-P; 75; see S3 Table] and is informed by the five stage framework for scoping reviews described by Arksey and O'Malley [76]. The five stages are: (1) identifying the research question, (2) identifying relevant studies, (3) study selection, (4) charting the data and (5) collating, summarising and reporting the results. The protocol for this scoping review has been registered with the Open Science Framework [77].

### Stage 1: Identifying the research question

Research questions were developed in adherence to the population, context, concept (PCC) framework as recommended by JBI [61; see Table 2]. The primary research question is as follows:

How has wellbeing been conceptualised in work on nature-based interventions?

Additional questions to guide the review are presented below:

1. What are the theoretical frameworks that underpin the published research in this area?

2. What types of nature-based interventions have been delivered?

3. What populations have been studied?

4. What outcomes (e.g., psychological, social, environmental) have been measured?

5. How and when does wellbeing arise through nature-based interventions?

**Table 1. Proposed timeline for scoping review.**

| Task | Month | Researcher(s) |
|---|---|---|
| Identify research question and define protocol | 1 | JB, AI, LJ, AK, AP |
| Study selection | 2–3 | JB, AP |
| Data Charting | 4–5 | JB, AP |
| Collating, summarising, reporting results | 5–6 | JB AP |
| Consultation exercise | 7–8 | JB, AP, AI, AK, LJ |
| Write-up of scoping review | 9–10 | JB |
| Review and Revision | 11–12 | JB, AI, LJ, AK, AP |

**Table 2. The population, concept, and context relating to the primary research question.**

| Population | Concept | Context |
|---|---|---|
| Individuals exposed to a nature-based intervention | Interventions in which the target population has been deliberately exposed to nature with the purpose of enhancing a measure of wellbeing, broadly defined, spanning the individual, others and planet | The context of the scoping review will remain 'open' |

## Stage 2: Identifying relevant studies eligibility criteria

**Search strategy.**   The search strategy was informed by JBI guidelines [61,62] and the methodology outlined by Arksey and O'Malley [76]. To ensure a comprehensive list of search terms, the authors completed several consultation exercises to identify terms related to the key themes of wellbeing and NBIs. Development of the search term list proceeded in two phases. Firstly, an initial set of search terms was generated from existing literature by drawing upon the research team's diverse expertise, including clinical practice (LJ), wellbeing science (AK), and sustainability and wellbeing research (AI). During the first consultation exercise, authors convened to exchange search terms. Discussion during this consultation stimulated additions to the list. In the second phase, which focused on broadening the list, search terms were inputted into the four target databases identified as relevant to this study (see 'Search Strategy'), and the resulting publications were 'term farmed.' The list was expanded again by inputting each search term into APAPsycInfo's Thesaurus of Psychological Index Terms tool, with the 'explode' function enabled. A second consultation allowed authors to discuss the expanded list, leading to additional term suggestions from the team. Search terms which repeatedly produced studies outside of the scope of this review were discarded or modified with prefixes or suffixes at this stage. For example, the term 'purpose,' chose for its relation to individual wellbeing, resulted in a large number of unrelated papers due the common phrase 'the purpose of this study.' The complete search strategy can be seen in Table 3.

Following consultation with experienced librarians, the following databases were identified as suitable for a search of the full search term list: APAPsycINFO, Web of Science, Medline, and Scopus. The ProQuest Dissertations & Theses Global database will be used to identify relevant unpublished research dissertations. Database-specific search strategies using the identified search terms are displayed in S2 Appendix. Any alterations made to search strategies during the review process will be reported in the full review. Data will be stored and screened using EndNote 20.6 [78] and Covidence systematic review software [79].

## Eligibility criteria

**Population.**   An aim of the present scoping review is to assess the methodologies through which NBIs have been delivered. This includes assessing the type of population(s) upon which NBIs have been investigated. Accordingly, studies will not be excluded based upon any participant characteristics.

**Concept.**   The present scoping review relates to the concept of 'nature-based interventions' (NBIs). Thus, studies must include the deliberate exposure of patients to natural environments – including natural environments self-identified by the authors of the study, or otherwise by the authors of the present review – with an outcome measure related to the concept of wellbeing (see Table 3). For the purpose of this scoping review, exposure may include real or virtual exposure to natural environments. Animal-assisted therapies will be included only if they involve exposure to a natural environment, such as equine-assisted therapies in woodland settings. Animal-assisted therapies conducted indoors, such as dog-assisted therapies in a hospital setting, will not be included, as they do not meet the criteria of natural environment exposure.

The concept of wellbeing presented in the current scoping review is informed by social ecological models of wellbeing, such as the GENIAL framework [11,34], and thus recognises the breadth of domains across which wellbeing spans. This includes measures ranging from the immediate wellbeing of an individual (e.g., self-reported wellbeing), to community-related measures (e.g., community cohesion), and measures related to the health of planetary-spanning ecosystems (e.g., pro-environmental attitudes). The domains of wellbeing and their related concepts identified as relevant in the present review may be seen in Table 3.

**Table 3. Research concepts and associated search terms.**

| Concept | Search term | Variations |
|---|---|---|
| Nature-based interventions | Nature-based therapy/therapies | Nature-based therapy/therapies<br>Nature-based intervention(s)<br>Nature-based activity/activities<br>Nature therapy/therapies<br>Nature exposure<br>Green therapy/therapies<br>Green space(s)<br>Blue therapy/therapies<br>Blue space(s)<br>Brown therapy/therapies<br>Brown Space(s)<br>Ecotherapy/therapies<br>Outdoor therapy/therapies<br>Outdoor healthcare<br>Outdoor behavioural healthcare<br>Environmental therapy/therapies<br>Green prescription(s)<br>Green gym(s)<br>Nature walk(s)<br>Simulated natural environment(s)<br>Animal-assisted therapy/therapies<br>Animal-assisted intervention(s)<br>Animal-assisted activity/activities<br>Pet therapy/therapies<br>Dog therapy/therapies<br>Canine therapy/therapies<br>Horse therapy<br>Equine therapy/therapies<br>Emotional support animal(s)<br>Emotional support pet(s) |
| | Forest-based therapy/therapies | Forest therapy/therapies<br>Forest bathing<br>Shinrin-yoku |
| | Horticultural Therapy | Horticulture therapy/therapies<br>Horticulture intervention(s)<br>Garden therapy/therapies<br>Garden intervention(s)<br>Wellbeing garden(s)<br>Rehabilitation garden(s)<br>Care farm(s)<br>Community farms(s) |
| | Wilderness therapy/therapies | Wilderness-based therapy/therapies<br>Wilderness-adventure therapy/therapies |
| | Adventure therapy/therapies | Adventure-based therapy/therapies<br>Adventure education<br>Therapeutic adventure(s) |

*(Continued)*

**Table 3.** (Continued)

| Concept | Search term | Variations |
|---|---|---|
| Wellbeing | Individual wellbeing | Wellbeing |
| | | Acceptance |
| | | Autonomy |
| | | Self-determination |
| | | Hope |
| | | Optimism |
| | | Humour |
| | | Spirituality |
| | | Self-efficiency |
| | | Self-esteem |
| | | Self-identity |
| | | Resilience |
| | | Intimacy |
| | | Psychological health |
| | | Mental health |
| | | Depression |
| | | Anxiety/Anxieties |
| | | Stress |
| | | Distress |
| | | Mental illness(es) |
| | | Psychopathology |
| | | Life satisfaction |
| | | Quality of Life |
| | | Positive affect |
| | | Negative affect |
| | | Physical health |
| | | Heart rate variability |
| | | Cortisol |
| | | Positive Emotion(s) |
| | | Negative Emotion(s) |
| | | Hedonic |
| | | Eudaemonia |
| | | Happiness |
| | | Disability/Disabilities |
| | | Functioning |
| | | Engagement |
| | | Positive relationship(s) |
| | | Meaning |
| | | Accomplishment(s) |
| | | Achievement(s) |
| | | Restoration |
| | | Sense of purpose |
| | Collective wellbeing | Social wellbeing |
| | | Social capital |
| | | Social participation |
| | | Psychosocial |
| | | Collective |
| | | Loneliness |
| | | Affiliation |
| | | Connection(s) |
| | | Neighbourhood(s) |
| | | Community/Communities |
| | | Cohesion |
| | | Belonging |
| | | Benevolence |
| | | Relatedness |

*(Continued)*

**Table 3.** (Continued)

| Concept | Search term | Variations |
|---|---|---|
| | Planetary wellbeing | Environmental attitude(s) |
| | | Environmental concern(s) |
| | | Environmental belief(s) |
| | | Environmental behaviour(s) |
| | | Ecological attitude(s) |
| | | Ecological concern(s) |
| | | Ecological belief(s |
| | | Ecological behaviour(s) |
| | | Climate attitude(s) |
| | | Climate concern(s) |
| | | Climate belief(s) |
| | | Climate behaviour(s) |
| | | Nature Connectedness |
| | | Nature relatedness |
| | | Nature ambivalence |
| | | Nature behaviour(s) |
| | | Pro-environmental |
| | | Environmental stewardship |
| | | Environmental value(s) |
| | | Biospheric value(s) |
| | | Self-transcendent value(s) |
| | | Materialistic value(s) |
| | | Biophilia |
| | | Biophobia |
| | | Human Value(s) |
| | | Personal Value(s) |
| | | Universalism |

**Context.** Context will remain 'open' for the present study [61]. Thus, studies presenting primary data (i.e., original research) will be included regardless of contextual factors, such as time of publication, geographic location, or healthcare population.

**Types of sources.** To be included in the review, studies must be published in English language with full text available to authors. To ensure a focus on validated sources of evidence, we have opted to exclude grey literature, focusing instead on theses and peer-reviewed journal articles. Whilst this may limit the diversity of evidence captured, this approach prioritises reliability and consistency. Future research could incorporate grey literature to explore additional perspectives and practice-based insights. In the absence of institutional access to the full text, the corresponding author of the study will be contacted via email to request a copy of their work, with a maximum of three attempts. No restrictions will apply to methodology or study design. The review will include randomised control trials, quasi-experimental designs, focus groups, and interviews. Additional study designs included during the review process will be reported in full in the review. Inclusion and exclusion criteria are presented in Table 4.

## Stage 3: Study selection

Duplicates will be removed and, of the remaining texts, titles and abstracts will undergo a pilot screening. In line with JBI methodology, this pilot exercise will consist of each named author reviewing the title and abstracts of a random sample of 25 papers. Authors will convene to compare included and excluded papers. A minimum consensus 75% will be required before the full screening proceeds. Adjustments to the criteria will be made iteratively as required and disclosed along with any other amendments in the methodology of the full review. Two

**Table 4. Inclusion and exclusion criteria.**

| Criterion | Inclusion | Exclusion |
|---|---|---|
| Type of study | Primary research evidence reporting on wellbeing-based outcomes targeted by a planned NBI. This includes randomised control trials, quasi-experimental designs, focus groups, and interviews | Study does not have a wellbeing-based outcome targeted by planned NBI. This will include review papers, meta-analyses, case-studies, and non-intervention-based research |
| Source | Study located in peer-reviewed journal or available as an e-thesis | Study not in peer-reviewed journal (e.g., grey literature) or available as an e-thesis |
| Availability | Full text available | No full text available |
| Study focus | Use of at least one nature-based intervention | Study is not conducted within the context of a nature-based setting |
| Language | English language | Non-English language |

reviewers, JB and AP, will apply the final eligibility criteria to a review of all study titles and abstracts. For each study, each reviewer will independently vote 'yes, include,' 'no, exclude,' or 'maybe, unknown.' Studies receiving a unanimous 'no' will be excluded and studies receiving a unanimous 'yes' will be included. The inclusion or exclusion of studies with a unanimous 'maybe' or without a unanimous result will be determined following a discussion between the two reviewers. A third reviewer will offer input to break ties when needed. Following the recommendations of Tricco et al. [74], a PRISMA flow diagram will be generated through Covidence to record the number of studies included and excluded – and the rationale of these decisions – at each stage of the study selection process. The proposed flow of articles through this process can be seen in Fig 1.

## Stage 4: Charting the data

Data will be extracted by two independent reviewers (JB and AP) with the use of a data charting form (see Table 5). Data extraction will be piloted by JB and AP on a randomly selected sample of 10 papers prior to a calibration exercise with the remaining authors (AI, LJ, AK). The calibrated form will be applied to all papers retained following screening. Data extraction will involve regular discussion between the two reviewers; disagreements will be resolved by input from a third reviewer. Data to be extracted will include general study information (e.g., title, first author name, year, population demographics, key findings) and the concepts (e.g., wellbeing outcomes and characteristics of NBI) and context (targeted population, geographic location) identified in the present review. NBI categorisation will broadly follow the five-category grouping presented by Harper et al. [25] but will remain an iterative process. Changes made to the categorisation system will be disclosed in the review. In the instance that the self-identification of an NBI is not recorded in a study or the authors' self-identification of the NBI does not align with Harper et al.'s categories [25], a note describing the rationale of subsequent categorisation will be provided by the authors in the full review. Any information from the data charting form which is missing from the study will be coded as 'not recorded,' and unclear information as 'unclear.' The reviewers will attempt a maximum of three email contacts with the study's corresponding author to request information or clarification. If the information remains inaccessible or unclear, this will be reported in the final review. Following data charting, a final consultation exercise with all authors will be held to review the suitability of the planned data analysis and presentation strategy, providing an opportunity to resolve unexpected findings. For example, review of the literature may reveal a sixth type of NBI not covered by Harper et al.'s [25] classification. In this instance, the authors will be able

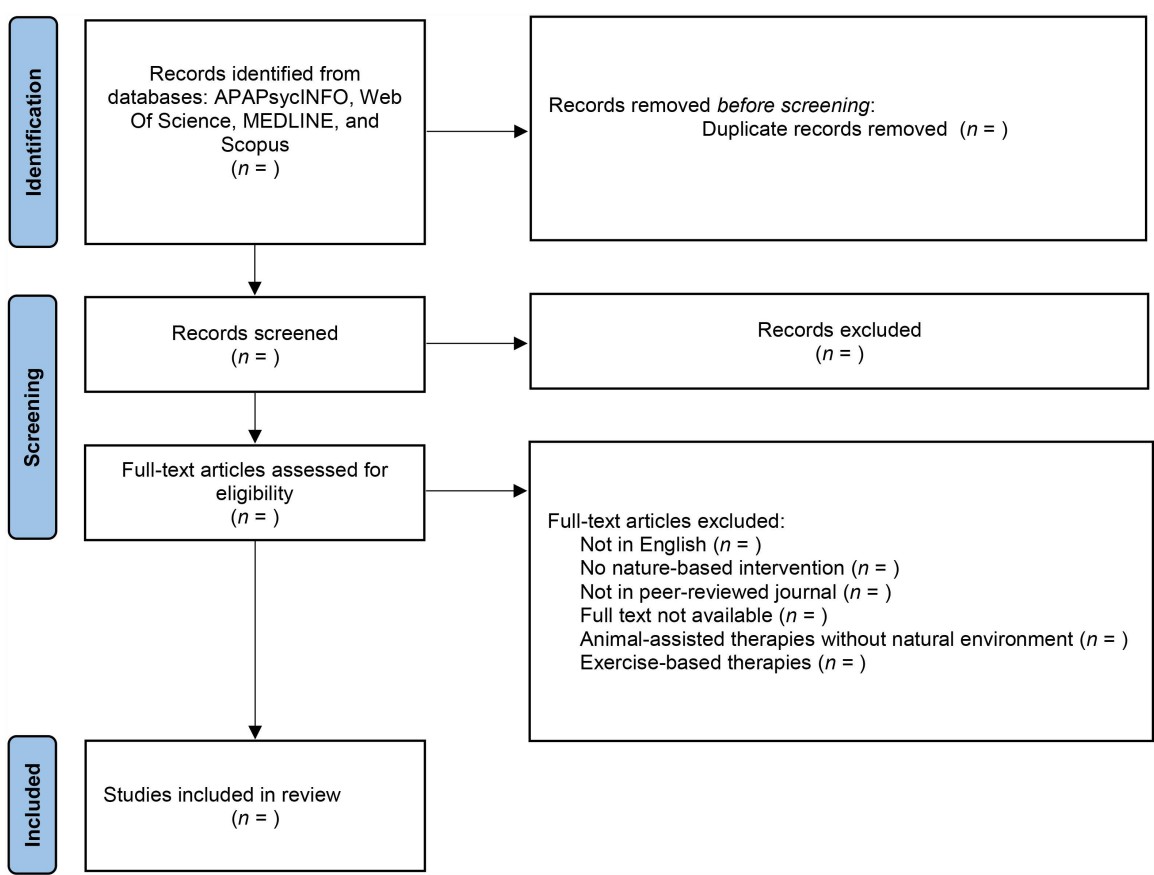

**Fig 1. Proposed flow of articles through the scoping review process.**

to amend the planned data presentation strategy. The details of any such amendments and their rationale will be reported in full in the review. Finally, given that the purpose of scoping reviews is to map the breadth of existing evidence rather than to assess methodological quality, and due to the typically heterogeneous nature of included sources, risk of bias assessments are generally considered unsuitable for scoping reviews [62,74] and will not be included in the present review.

## Stage 5: Collating, summarising and reporting the results

Following data charting, a final consultation exercise with all authors will be conducted to develop a data analysis and presentation strategy. The details of this data analysis and presentation will be reported in full in the review. Expected data presentation related to each research question may be seen in Table 5. All data will be published in the full review. Data synthesis and presentation will be informed by the multi-domain approach to wellbeing imposed by the GENIAL framework, with several sub-group analyses to handle the expected heterogeneity of results. Quantitative synthesis will involve tabular and graphical summaries showing the frequency of studies that report data on each facet listed in Table 5 (a sample of one such table is presented as Table 6 below). This synthesis will highlight any imbalances in the distribution of studies across methodological approach (quantitative or qualitative), NBI type, theoretical framework (e.g., Attention Restoration Theory), psychotherapeutic approach (where relevant; e.g., Cognitive Behavioural Therapy), geographic location, population

**Table 5. Data charting form.**

| Facet | Data |
|---|---|
| Publication details | Title<br>First Author<br>Year<br>Location of study (country)<br>Field of study, first author |
| Target population | Record details of targeted population |
| Study demographics | Sample size<br>Age range<br>Mean age<br>Sex<br>Gender identity<br>Nationality<br>Ethnicity |
| Study Design | Qualitative<br>Quantitative (randomised control trial; quasi-experimental)<br>Mixed method<br>Recruitment method and source of participant referral (where applicable) |
| Nature-based intervention characteristics | Type of nature-based intervention (as termed by studies' authors) Orientation of psychotherapeutic approach (where relevant; e.g., cognitive behavioural therapy) Categorisation of intervention into NBI-type derived from Harper et al.'s [25] review. NBI details:•<br>Setting (e.g., local park, health care setting, university setting)<br>• Timeframe of intervention<br>• Duration of contact<br>• Frequency of contact<br>• Type of interaction (as *per* Keniger et al.'s [80] typology of interactions)<br>• Description of nature (where applicable)<br>• Mode of delivery (e.g., facilitated vs. unfacilitated) and theoretical approach (e.g., mindfulness) |
| Outcomes measured | Record outcomes from study (including effect sizes where applicable)<br>Categorise outcomes by wellbeing domain<br>Record presence of moderators and mediators (including significance levels where relevant) |
| Findings | Description of key findings including duration of follow-up measures |

(clinical or non-clinical), and outcome by wellbeing domain, ultimately pointing to areas for future research.

The expected heterogeneity of results from quantitative and qualitative methods precludes a direct statistically comparison of findings. However, a broad synthesis of heterogenous results can be achieved through narrative synthesis and dual-display tables. A narrative synthesis will examine how wellbeing is conceptualised and measured in NBI studies, reflecting on proposed mechanisms of change and the extent to which NBI research adopts multi-faceted conceptualisations of wellbeing. Dual-display tables, which provide a structured presentation of key findings across methodologies, will be used to synthesise quantitative and qualitative findings across the multiple domains of wellbeing as *per* the GENIAL framework (for a sample dual-display table, see Table 7). In summary, data synthesis will answer the research questions proposed in Stage 1, by presenting an overview of

the following findings: (1) definitions and conceptualisations of wellbeing, (2) theoretical frameworks utilised, (3) key characteristics of NBIs, (4) how wellbeing is measured, and (5) populations that have been studied. The narrative synthesis will also identify any proposed mechanisms of change reported in the literature, and highlight any commonalities in terms of the specific domain of wellbeing to which the outcomes relate.

## Conclusion

In this paper, we present a protocol for a scoping review that will provide a comprehensive overview of primary research on NBIs designed to enhance wellbeing. This scoping review will impose a holistic framework for conceptualising wellbeing on the heterogeneous evidence base relating to the impact of NBIs on wellbeing. The synthesis will enhance understanding of

**Table 6. Planned table of outcome measures and direction of change reported by quantitative investigations of NBIs by wellbeing domain.**

| Wellbeing outcome | NBI | Forest intervention | Horticultural intervention | Wilderness intervention | Adventure intervention |
|---|---|---|---|---|---|
| **Individual** | | | | | |
| *Measure 1* | | | | | |
| *Measure 2* | | | | | |
| *Measure 3* | | | | | |
| **Collective** | | | | | |
| *Measure 1* | | | | | |
| *Measure 2* | | | | | |
| *Measure 3* | | | | | |
| **Planetary** | | | | | |
| *Measure 1* | | | | | |
| *Measure 2* | | | | | |
| *Measure 3* | | | | | |
| ***n* Studies** | | | | | |

*NB. Each circle represents one study:* ● = positive outcome, ◑ = mixed outcome or no significant change, ○ = negative outcome.

**Table 7. Planned dual-display table of the key findings from quantitative and qualitative investigations of NBIs by wellbeing domain.**

| Wellbeing outcome | Quantitative results | Qualitative results |
|---|---|---|
| **Individual** | | |
| *Key finding 1* | | |
| *Key finding 2* | | |
| *Key finding 3* | | |
| **Collective** | | |
| *Key finding 1* | | |
| *Key finding 2* | | |
| *Key finding 3* | | |
| **Planetary** | | |
| *Key finding 1* | | |
| *Key finding 2* | | |
| *Key finding 3* | | |

how different types of NBIs impact on individual, collective and planetary wellbeing, promoting a more integrated perspective and supporting policy makers and practitioners to make more informed decisions. This review will also support the development of wellbeing-focused interventions with a focus on better managing the intersecting challenges associated with the unfolding climate, biodiversity and ecological crises, alongside the loneliness epidemic and increasing burden of non-communicable disease.

## Supporting information

**S1 Appendix. Search strategy.**
(DOCX)

**S2 Table. Database-specific search strategies.**
(DOCX)

**S3 Table. PRISMA-P 2015 checklist.**
(DOCX)

## Acknowledgments

The database selection to be used in this scoping review was developed with the assistance of several experienced librarians from Swansea University. The authors would like to thank Stephen Storey, Giles Lloyd-Brown, and Elen Davies for their contributions. An additional thanks is extended to Elen Davies for her support in advising database-specific search strategies.

## Author contributions

**Conceptualization:** Jesse Blackburn, Afonso Pereira, Luke Jefferies, Andrew H. Kemp, Amy Isham.

**Methodology:** Jesse Blackburn, Afonso Pereira, Luke Jefferies, Andrew H. Kemp, Amy Isham.

**Project administration:** Jesse Blackburn, Luke Jefferies, Andrew H. Kemp, Amy Isham.

**Supervision:** Luke Jefferies, Andrew H. Kemp, Amy Isham.

**Writing – original draft:** Jesse Blackburn, Andrew H. Kemp.

**Writing – review & editing:** Afonso Pereira, Luke Jefferies, Amy Isham.

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
