## [Decision Letter · Decision Letter 0]

11 Sep 2024

PONE-D-24-32661Nature-based interventions for individual, collective and planetary wellbeing: A protocol for a scoping reviewPLOS ONE

Dear Dr. Isham,

Thank you for submitting your manuscript to PLOS ONE. After careful consideration, we feel that it has merit but does not fully meet PLOS ONE’s publication criteria as it currently stands. Therefore, we invite you to submit a revised version of the manuscript that addresses the points raised during the review process.

We look forward to receiving your revised manuscript.

Kind regards,

Cho-Hao Howard Lee, M.D.

Academic Editor

PLOS ONE

Journal Requirements:

2. We note that you have referenced (Pereira, Blackburn et al., unpublished) on page 6, which has currently not yet been accepted for publication. Please remove this from your References and amend this to state in the body of your manuscript: (ie “Bewick et al. [Unpublished]”) as detailed online in our guide for authors

3.Please review your reference list to ensure that it is complete and correct. If you have cited papers that have been retracted, please include the rationale for doing so in the manuscript text, or remove these references and replace them with relevant current references. Any changes to the reference list should be mentioned in the rebuttal letter that accompanies your revised manuscript. If you need to cite a retracted article, indicate the article’s retracted status in the References list and also include a citation and full reference for the retraction notice.

Reviewers' comments:

Reviewer's Responses to Questions

**Comments to the Author**

1. Does the manuscript provide a valid rationale for the proposed study, with clearly identified and justified research questions?

Reviewer #1: Yes

Reviewer #2: Yes

2. Is the protocol technically sound and planned in a manner that will lead to a meaningful outcome and allow testing the stated hypotheses?

Reviewer #1: Yes

Reviewer #2: Yes

3. Is the methodology feasible and described in sufficient detail to allow the work to be replicable?

Reviewer #1: Yes

Reviewer #2: Yes

4. Have the authors described where all data underlying the findings will be made available when the study is complete?

Reviewer #1: Yes

Reviewer #2: Yes

5. Is the manuscript presented in an intelligible fashion and written in standard English?

Reviewer #1: Yes

Reviewer #2: Yes

6. Review Comments to the Author

You may also provide optional suggestions and comments to authors that they might find helpful in planning their study.

Reviewer #1: This manuscript presents a protocol for a scoping review that seeks to investigate the impacts of nature-based interventions (NBIs) on individual, collective, and planetary wellbeing. The topic is highly relevant and timely, particularly given the growing interest in the intersections between human health, social wellbeing, and environmental sustainability. The study has the potential to contribute significantly to this emerging field, but there are several critical areas where the manuscript requires substantial improvement to enhance its clarity, comprehensiveness, and methodological rigor.

Major Issues:

1. The current search strategy, while it includes multiple databases, does not provide enough detail to ensure that the literature review will be comprehensive. The manuscript should elaborate on the development of the search terms, including how keywords were selected and refined. Moreover, it would be beneficial to include additional sources, such as grey literature, theses, and conference papers, to minimize publication bias.

2. The protocol does not adequately address how the expected heterogeneity among studies will be managed. Given the broad scope of the review, which includes various study designs, populations, and outcomes, it is crucial to outline a clear strategy for synthesizing different types of evidence. This could involve subgroup analyses, sensitivity analyses, or the use of narrative synthesis to handle variability in study designs and outcomes.

3. While the manuscript references several theoretical frameworks, including the GENIAL framework, it falls short in explaining how these frameworks will be applied in the analysis and interpretation of data. Theoretical frameworks are critical in guiding the synthesis of findings and ensuring that the review’s conclusions are grounded in a robust conceptual foundation. The authors should provide more detailed explanations of how these frameworks will be operationalized in the review process. This would significantly enhance the theoretical rigor of the study and provide a clearer roadmap for interpreting the results.

4. The manuscript discusses individual, collective, and planetary wellbeing, but the integration of these concepts into a coherent framework is somewhat lacking. It is important to clarify how these dimensions of wellbeing will be assessed and integrated within the context of NBIs. A more detailed conceptual framework that explicitly connects these dimensions would help to unify the review and make its contributions more substantial.

Minor Issues:

1. The manuscript currently limits the scope of the literature review to English-language studies. However, some published reviews in this field have included studies in multiple languages, such as Portuguese, French, English, and Spanish. If possible, expanding the language scope of the review to include these and potentially other languages could enhance the comprehensiveness of the review. By incorporating studies in additional languages, the authors could potentially uncover relevant research that might be overlooked due to language barriers, thereby providing a more global perspective on the impact of nature-based interventions on wellbeing.

2. The manuscript contains minor inconsistencies in terminology, such as the alternation between "wellbeing" and "well-being."

Reviewer #2: I think this manuscript presents an exciting and much-needed framework for studying Nature-Based Interventions (NBIs) with a holistic view. It stands out because it goes beyond traditional reviews by incorporating a focus on collective and planetary wellbeing, which hasn’t received enough attention in previous work. The methodology is sound, well-structured, and adheres to well-recognized frameworks such as the Joanna Briggs Institute (JBI) and PRISMA guidelines. However, there are a few areas where the manuscript could be more transparent and detailed to maximize its impact.

Strengths:

Novel Scope and Holistic Approach: The focus on a tri-dimensional view of wellbeing—individual, collective, and planetary—is a significant advancement in the study of NBIs. I really appreciate the expansion of the concept of wellbeing beyond individual health. It’s clear that the authors aim to break away from the narrow, individualistic focus that dominates current literature, which is refreshing and forward-thinking.

Clear and Rigorous Methodology: The authors have established a detailed methodology, with clear adherence to established protocols. The inclusion of multiple types of studies (RCTs, quasi-experimental designs, interviews) is a positive aspect of the review, making the protocol robust and inclusive. Using the Covidence platform for data screening also adds an extra layer of rigor and transparency, which I find commendable.

Addressing Knowledge Gaps: One major strength of this protocol is that it directly addresses the gaps in the existing literature, particularly the omission of social and planetary wellbeing in previous reviews. The use of a more diverse set of search terms—like community cohesion, environmental attitudes, and pro-environmental behaviors—shows that the authors have really thought through how to capture these neglected dimensions.

Consultation Exercise: Including a consultation phase with stakeholders and experts adds value to the review, ensuring that the outcomes are both academically sound and practically relevant. This is an innovative addition that I believe will increase the review's credibility and usefulness for policymakers and practitioners.

Areas for Improvement:

Search Strategy Clarification: The inclusion of multiple search terms for various wellbeing domains is a great idea, but I feel the strategy could benefit from some fine-tuning. For instance, there are several overlapping terms in the list (e.g., “environmental attitudes” vs. “ecological attitudes”), and I worry this could lead to redundancy or a dilution of focus. The authors could consider consolidating some of these terms to ensure the review remains focused and concise.

Handling Study Heterogeneity: Given the wide variety of studies that will be included, it’s likely that there will be considerable heterogeneity in terms of intervention types, outcomes measured, and populations studied. While the authors acknowledge this, I think they should be more explicit about how they will handle this during data synthesis. Will they consider subgroup analyses based on factors like study design or NBI type? Clarifying this would help the reader understand how the results will be interpreted in light of such variability.

Definitions of Key Concepts: Although the authors have done a great job introducing the tri-dimensional framework of wellbeing, I think it would help if they provided a bit more clarity around some of the terms. For example, “planetary wellbeing” is a relatively new concept and could be confusing for readers unfamiliar with this area. A clearer, more operational definition would improve the accessibility of the manuscript.

Potential for Bias: The authors plan to include studies from various methodologies, which is great for inclusivity. However, I think it would be beneficial if they were more upfront about how they will assess the risk of bias in these studies. Will there be any formal bias assessment tools applied, or is this not a focus for this review? Clarifying this point would make the review more robust.

Specific Suggestions:

Consultation Process: The authors mention a consultation exercise but don’t provide enough detail on how it will be conducted. I think it would be helpful to know how the feedback from the consultation will be integrated into the final review. Will it simply validate the findings, or could it result in changes to the data synthesis? A bit more transparency here would be useful.

Timelines and Feasibility: The proposed timeline (Table 1) looks ambitious. Completing the entire review, including data extraction, collation, and the consultation exercise, in just six months might be too optimistic, particularly if they encounter unexpected challenges. The authors may want to revise the timeline to reflect a more realistic completion date, or provide a contingency plan in case of delays.

Data Presentation: I feel the authors could give more detail on how they plan to present the data, especially across the three domains of wellbeing (individual, collective, planetary). Will these be analyzed separately or together? It’s important to know this, as it will impact the interpretation of the findings.

Conclusion:

This protocol is well-written and tackles a very relevant and underexplored area. With minor adjustments—particularly around the search strategy, handling heterogeneity, and a more detailed consultation process—this protocol will be ready for publication. I believe it will make a valuable contribution to the literature on NBIs, particularly with its novel focus on collective and planetary wellbeing.

7. PLOS authors have the option to publish the peer review history of their article (what does this mean? ). If published, this will include your full peer review and any attached files.

**Do you want your identity to be public for this peer review?** For information about this choice, including consent withdrawal, please see our Privacy Policy .

Reviewer #1: No

Reviewer #2: **Yes: ** Xiaoyi Zhang, MD

---

## [Author Response · Author response to Decision Letter 1]

17 Oct 2024

Reviewer #1

Opening comment

We are grateful to the reviewer for their thoughtful and constructive feedback. We appreciate the recognition of our work’s relevance and potential contribution to the field. We believe the revisions have helped to improve the manuscript, and we hope that we have addressed the reviewer’s suggestions in a way that strengthens the overall quality and clarity of the protocol.

Point 1

Thank you for this suggestion. We have provided a more detailed description of this process across lines 232-246.

In light of this comment, and in recognition of the need to reduce publication bias, we have also chosen to extend our search to include online repositories of unpublished theses and dissertations. This is reflected on lines 32-34, 251, 252, 286 and in Table 4.

However, due to the large number of search terms and inclusion of four databases, we expect a large output of published material. Indeed, a preliminary search using the search term list returned a very large number of papers (7500+). Given the quantity of peer-reviewed literature we expect to capture, we have chosen not to extend our search to grey literature. Instead, we have chosen to focus on the most validated sources of evidence. Finally, we have opted not to include conference papers as these are commonly later published in peer review sources.

Point 2

We have responded to this comment on lines 252-272, where we now outline a description of the intended methods of synthesising review outcomes.

In summary, the data extraction form has been designed to capture the heterogeneity of data across studies; and this data will be mapped in tabular form.

In addition, we intend to complete several sub-group analyses. Studies will be grouped by methodological approach (quantitative or qualitative) and wellbeing domain, as outlined by the GENIAL framework. Outcomes will be charted based on which subgroup they fall into.

Discussion of findings will also be grouped by:

• NBI type

• Theoretical frameworks cited (e.g., stress-reduction theory)

• Psychotherapeutic approach

• Geographic location

• Population (clinical vs. non-clinical)

Finally, a full data charting table will be available as supplementary information.

For each study, this table will detail:

• Publication details

• Population and sample details

• Study details (methodology)

• Outcomes

Any alterations to this plan will be disclosed in the full report. Additionally, we have now outlined in the paper how a group consultation exercise will be held following data extraction to ensure intended reporting methods are still appropriate given the nature of review findings (lines 235-240).

Point 3

In response to this comment, the data presentation plan across lines 352-372 has been updated.

In summary, the frameworks referred to in the manuscript, including GENIAL, impose a multi-domain framework of wellbeing. Accordingly, data will be displayed with reference to these frameworks.

This will include a narrative synthesis addressing the extent to which results span across the domains of wellbeing associated with the GENIAL model (i.e., individual, collective, and planetary wellbeing), and will inform the use of data display tables. For example, quantitative and qualitative findings will be presented in a single dual-display table by wellbeing domain with the following column and row headings:

Column headings:

• Wellbeing Domain

• Quantitative Findings

• Qualitative Findings

Row headings:

• Individual wellbeing

• Collective wellbeing

• Planetary wellbeing

The inherent heterogeneity of the results will require summarisation within the dual-display table. Quantitative data will be identified with reference to the assessment scale used to collect data and the direction of change (positive, negative, or no change). Qualitative data will be condensed into brief text summaries, highlighting key themes and patterns observed.

This approach will facilitate a clear comparison of results across studies while accommodating the diverse nature of the findings. A narrative synthesis will then summarise the nature and consistency of results stemming from quantitative and qualitative approaches to data collection.

The GENIAL framework will also be used to display the distribution of studies reporting outcomes by wellbeing domain.

The frameworks referred to in the manuscript, including GENIAL, impose a multi-domain framework of wellbeing. Accordingly, data will be displayed with reference to this framework. One such data display will consist of a Venn diagram depicting the distribution of outcomes by wellbeing domain. This figure will synthesise quantitative and qualitative findings into a single display.

An example of the Venn diagram may be seen in the 'Response to Reviewers' document.

Point 4

In response to this comment, our description of the GENIAL framework has been added to lines 111-118 and 140-142.

It is also our intention that the updated data presentation plan across lines 352-372 outlines the suitability of the frameworks discussed in the manuscript, including GENIAL, for conceptualising and assessing the reciprocal and cumulative effects of promoting wellbeing at multiple levels through NBIs.

Point 5

We recognise this limitation and this will be noted in the final report. This study is being completed as part of time-limited PhD project. Unfortunately, as a result, we have neither sufficient time nor access to the resources required to translate non-English language studies.

Point 6

After careful review of the manuscript, we have identified 28 uses of the term ‘well-being’ in this manuscript. However, the use of the term ‘well-being’ is limited to (1) the reference list, where ‘well-being’ has been used to preserve the original titles of the referenced works, and (2) in the search item list where every instance of the word ‘wellbeing’ has been repeated as ‘well-being’ and ‘well being’ to reflect the varied usage of the term across the literature. Outside of these contexts, we have checked that we are consistent in the use of 'wellbeing'.

Reviewer #2

Opening comment

Point 1

We thank the reviewer for their kind words and thorough review of our manuscript. We hope that we have addressed the reviewer’s suggestions, and certainly recognise these comments as contributing to a stronger manuscript.

Point 2

We thank the review for their thoughtful and encouraging feedback. It is indeed our intention to expand the concept of wellbeing beyond the traditional individualistic approach that has dominated current literature. We believe that a holistic perspective is essential for fully understanding the multifaceted benefits of nature-based interventions.

Point 3

We are grateful for your support and look forward to incorporating your insights as we refine our manuscript.

Point 4

We thank the reviewer for their comments on the consultation exercise. As in our response to Reviewer #2, comment 9 below, we have amended the manuscript to provide more detail about each researcher’s area of expertise. This amendment may be seen on lines 232-246.

Point 5

The range of search terms was selected to capture the varied terminology used across disciplines. This breadth helps ensure comprehensive coverage of relevant studies. However, we acknowledge the concern about overlaps and redundancy. Any duplications of studies arising from these broad search terms will be removed before data charting commences.

Point 6

We acknowledge the difficulties of handling the heterogeneity of data arising from our methodology, and we have addressed this comment in lines 352-372 of the manuscript.

In summary, the frameworks referred to in the manuscript, including GENIAL, impose a multi-domain framework of wellbeing. Our data charting method has been designed to extract the extent to which studies on NBIs consider outcomes relating to multiple wellbeing domains identified by these frameworks (i.e., individual, collective, and planetary wellbeing). Data across all studies will, therefore, be synthesised by wellbeing domain.

To manage study heterogeneity, we will include several sub-group analyses (as above).

Sub-group analyses will include:

• Methodological approach (quantitative or qualitative)

• NBI type

• Theoretical frameworks cited (e.g., stress-reduction theory)

• Psychotherapeutic approach

• Geographic location

• Population (clinical vs. non-clinical)

The nature and consistency of findings across subgroups will then be discussed in a narrative synthesis.

Finally, a full data charting table will be available as supplementary information.

Point 7

We have endeavoured to respond to this feedback by elaborating on the definition of planetary wellbeing across lines 104-108, as quoted below:

“Planetary wellbeing can be considered a systems-oriented perspective that integrates ecosystem health and human wellbeing. The JYU.Wisdom Community [38] considers planetary wellbeing to be a state in which the Earth’s natural systems are able to sustain life whilst simultaneously fulfilling organisms’ needs, allowing them to actualise their inherent capacities.”

Point 8

Previous guidance for systematic scoping reviews has outlined risk of bias analysis as unsuitable or otherwise not recommended (Peters et al., 2020; Tricco et al., 2018). However, we agree that clarifying this point in the manuscript would be beneficial. This change has been implemented across lines 341-344. This reads:

“Finally, given that the purpose of scoping reviews is to map the breadth of existing evidence rather than to assess methodological quality, and due to the typically heterogeneous nature of included sources, risk of bias assessments are generally considered unsuitable for scoping reviews [62, 74] and will not be included in the present review.

References:

62 Peters, Marnie C, Tricco AC, Pollock D, Munn Z, Alexander L, et al. Updated methodological guidance for the conduct of scoping reviews. JBI Evid Synth. 2020;18(10):2119-26.

74 Tricco AC, Lillie E, Zarin W, O'Brien KK, Colquhoun H, Levac D, et al. PRISMA extension for scoping reviews (PRISMA-ScR): checklist and explanation. Annals of Internal Medicine. 2018;169(7):467-73.

Point 9

We have responded to this by adding the below content to the manuscript (lines 335-344):

“Following data charting, a final consultation exercise with all authors will be held to review the suitability of the planned data analysis and presentation strategy, providing an opportunity to resolve unexpected findings. For example, review of the literature may reveal a sixth type of NBI not covered by Harper et al.’s [25] classification. In this instance, the authors will be able to amend the planned data presentation strategy. The details of any such amendments and their rationale will be reported in full in the review.”

Point 10

The draft of the proposed timeline was created when researchers’ availability allowed full-time dedication to the review. However, in light of changing circumstances and the reviewer’s recommendation, the timeline of this project has been extended. This change is reflected in Table 1 and line 199.

In addition, the expected commencement of the study has been updated (line 200).

Point 11

In response to this comment, the data presentation plan across lines 352-372 has been updated.

In summary, we intend to complete several sub-group analyses. Studies will be grouped by methodological approach (quantitative or qualitative) and wellbeing domain, as outlined by the GENIAL framework. Outcomes will be charted based on which subgroup they fall into. several tables and graphs will display findings separately for the three domains of wellbeing. Briefly, this will include:

• Tables displaying outcomes, grouped by wellbeing domain, by intervention type

• Tables displaying the number of studies using each NBI type

• A display of the number of studies by intervention type and theoretical framework

• A table displaying reported population studies

• A heat map showing geographical distribution of studies

A narrative synthesis and dual-display table will then be used to integrate findings across methodological approach and wellbeing domains. This approach will reflect the interconnectedness of the wellbeing domains.

Finally, a full data charting table will be available as supplementary information.

For each study, this table will detail:

• Publication details

• Population and sample details

• Study details (methodology)

• Outcomes

Any alterations to this plan will be disclosed in the full report.

Concluding comment

We are grateful to the review for their insightful remarks and constructive feedback. We recognise that the reviewers comments have helped to strengthen our manuscript’s rigour and clarity.

---

## [Decision Letter · Decision Letter 1]

13 Nov 2024

Nature-based interventions for individual, collective and planetary wellbeing: A protocol for a scoping review

PONE-D-24-32661R1 

Dear Dr. Amy Isham,

We’re pleased to inform you that your manuscript has been judged scientifically suitable for publication and will be formally accepted for publication once it meets all outstanding technical requirements.

Kind regards,

Cho-Hao Howard Lee, M.D.

Academic Editor

PLOS ONE

Reviewers' comments:

Reviewer's Responses to Questions

**Comments to the Author**

1. Does the manuscript provide a valid rationale for the proposed study, with clearly identified and justified research questions?

Reviewer #1: Yes

Reviewer #2: Yes

2. Is the protocol technically sound and planned in a manner that will lead to a meaningful outcome and allow testing the stated hypotheses?

Reviewer #1: Yes

Reviewer #2: Yes

3. Is the methodology feasible and described in sufficient detail to allow the work to be replicable?

Reviewer #1: Yes

Reviewer #2: Yes

4. Have the authors described where all data underlying the findings will be made available when the study is complete?

Reviewer #1: Yes

Reviewer #2: Yes

5. Is the manuscript presented in an intelligible fashion and written in standard English?

Reviewer #1: Yes

Reviewer #2: Yes

6. Review Comments to the Author

You may also provide optional suggestions and comments to authors that they might find helpful in planning their study.

Reviewer #1: The authors' responses have addressed my concerns, and I support the publication of their work. Thanks. Good luck!

Reviewer #2: Thank you for the opportunity to review this revision. It’s clear that you’ve carefully addressed previous feedback, and the result is a strong, well-structured protocol. Your focus on exploring the impacts of nature-based interventions (NBIs) across individual, collective, and planetary wellbeing adds valuable breadth to the existing literature. This approach is both timely and impactful, and I believe your work will make a meaningful contribution to the field. Here are a few minor suggestions that could further enhance clarity and coherence.

Merits

1. Comprehensive Framework: The use of the GENIAL model to assess NBIs across multiple domains of wellbeing is a significant asset. It provides a holistic perspective, which will likely be of interest to a broad audience, from clinicians to policymakers.

2. Clear and Detailed Methodology: Your protocol is meticulously outlined, with transparent steps for database selection, inclusion/exclusion criteria, and data charting. This rigor enhances the replicability of your study and its potential for future use as a foundational work in the area.

3. Inclusive Approach to NBI Types: By including various forms of NBIs, from green and blue spaces to animal-assisted interventions, the protocol ensures a wide-ranging analysis, which will make the review’s findings broadly applicable.

Minor Suggestions for Revision

1. Clarify Definitions of Wellbeing Domains: Although you define the three domains of wellbeing, adding a brief example of each (e.g., individual wellbeing as mood improvement, collective wellbeing as community cohesion, and planetary wellbeing as pro-environmental behavior) could enhance reader understanding and make the distinctions even clearer.

2. Elaborate Slightly on Data Presentation for Each Domain: The data presentation plan is well-organized. Including a short description of how results will be grouped within each domain, perhaps through sample table headers or an illustrative example, could help readers visualize the final data structure.

3. Address the Exclusion of Grey Literature: Since this decision was made in the interest of focusing on validated sources, a brief acknowledgment of how this might impact the findings—alongside the reasoning you provided—would help address any lingering questions on potential publication bias.

4. Expand Briefly on the Consultation Exercise: The inclusion of a final consultation exercise is a great step for ensuring methodological soundness. A sentence or two on the types of expertise represented in this group would strengthen readers' understanding of how the consultation adds to the review's rigor.

5. Consider a Note on Expected Limitations Due to Heterogeneity: Given the broad scope of NBIs, you may encounter a high degree of heterogeneity. A brief note on how this will be handled, with an emphasis on narrative synthesis or alternative grouping methods if meta-analysis isn’t feasible, would set clear expectations for readers.

Recommendation: Minor Revision

Overall, this is a well-developed protocol that thoughtfully incorporates feedback from previous reviews. The minor revisions suggested are intended to further clarify and enhance the manuscript’s readability and impact. I am excited to see the completed review and the valuable insights it will bring to the field. Thank you for your hard work, and best of luck as you finalize this important study!

7. PLOS authors have the option to publish the peer review history of their article (what does this mean? ). If published, this will include your full peer review and any attached files.

**Do you want your identity to be public for this peer review?** For information about this choice, including consent withdrawal, please see our Privacy Policy .

Reviewer #1: No

Reviewer #2: **Yes: ** Xiaoyi Zhang, M.D.

---

## [Editor Report · Acceptance letter]

PONE-D-24-32661R1

PLOS ONE

Dear Dr. Isham,

I'm pleased to inform you that your manuscript has been deemed suitable for publication in PLOS ONE. Congratulations! Your manuscript is now being handed over to our production team.

Kind regards,

on behalf of

Dr. Cho-Hao Howard Lee

Academic Editor

PLOS ONE